# Oxygen Plasma Treatment of Rear Multilayered Graphene: A Potential Top Electrode for Transparent Organic Light-Emitting Diodes

**DOI:** 10.3390/ma14216652

**Published:** 2021-11-04

**Authors:** Jong Tae Lim

**Affiliations:** Reality Devices Research Division, Electronics and Telecommunications Research Institute, Daejeon 34129, Korea; lapbbi@etri.re.kr

**Keywords:** organic light-emitting diode, multilayered graphene, photoemission spectroscopy, near edge X-ray absorption fine structure spectroscopy

## Abstract

One of the core technologies of transparent organic light-emitting diodes (TOLEDs) is to develop an optically transparent and high electrical conductivity electrode so that light generated inside the device can efficiently escape into the air through the electrodes. We recently reported in TOLED research that two flipping processes are required to dry-transfer the front multilayered graphene (MLG) to the top electrode, while the rear MLG requires one dry transfer process. As the transfer process increases, the electrical properties of graphene deteriorate due to physical damage and contamination by impurities. At the charge-injecting layer/MLG interface constituting the TOLED, the rear MLG electrode has significantly lower charge injection characteristics than the front MLG electrode, so it is very important to improve the electrical characteristics of the rear MLG. In this paper, we report that the light-emitting properties of the TOLED are improved when an oxygen plasma-treated rear MLG is used as the top electrode, as compared with untreated rear MLG. In addition, the fabricated device exhibits a transmittance of 74–75% at the maximum electroluminescence wavelength, and the uniformity of transmittance and reflectance is more constant at a wavelength of 400–700 nm than in a device with a metal electrode. Finally, near-edge X-ray absorption fine structure spectroscopic analysis proves that the MLG crystallinity is improved with the removal of impurities on the surface after oxygen plasma treatment.

## 1. Introduction

Fragile tin-doped indium oxide (ITO) has been most commonly used as a transparent conducting electrode of organic light-emitting diodes (OLEDs) since the first report on light emission from such devices in 1987 [1]. With increasing demand for—and necessity of—flexible optoelectronic devices [2], great efforts have been made to replace ITO [3] with non-fragile materials such as graphene [4,5,6], semi-transparent thin metal layers [7,8], conductive polymers [9], carbon nanotubes [10], metal grids [11], and metallic nanowires [12,13]. Meanwhile, it has been demonstrated that a top-emission (TE) architecture makes it possible to design large aperture displays that can achieve high resolution, a high degree of design freedom of pixel circuits, and only a little loss of light due to a decrease in waveguide loss in a glass substrate. Such characteristics are superior to bottom-emission (BE) structure displays [14]. In addition, inverted TE OLEDs offer the advantage of using n-type amorphous silicon thin film transistors, which provide more uniform brightness for large area displays.

In inverted OLEDs, understanding of the interface between the top anode and the hole-injecting layer (HIL) is central to the device physics because the mismatch between the Fermi level of the anode and the highest occupied molecular orbital of HIL controls hole injection in the device. More specifically, a high work function anode, which reduces the barrier height for hole injection at the anode–HIL interface, is necessary to fabricate an efficient device [15]; thus, careful alignment of energy levels in the interface is necessary to maximize the luminous efficiency and minimize the power consumption. This is done by designing an excellent interface in which carriers are effectively injected at low voltage.

The use of the front face of multilayered graphene (MLG) has been recently understood as a new method to achieve efficient interfaces of inverted devices [16]. One device with an HIL–front MLG interface showed significantly improved light-emission properties compared to those of a device with an HIL–rear MLG interface: the improvement was due to easier hole injection and higher crystallinity on the front face of MLG. In addition, an etching-free ozone treatment has been proposed to tune the electrical resistance and the optical transmittance of graphene films by simply varying the time and temperature of graphene exposure to ozone [17]. Furthermore, compared to that of an untreated surface, extremely low contact resistances on graphene can be obtained through various modifications of graphene itself [18] as well as oxygen [19,20] and ozone [21] treatment.

Due to the growing importance of highly transparent display technology, demand for MLGs is greatly increasing. The front MLG surface, with few impurities and high crystallinity, can be used as a bottom electrode of a transparent OLED through a one-time transfer process. However, to apply the rear MLG surface, which has low crystallinity due to relatively many impurities, as the top electrode of a transparent OLED, two flipping transfer processes are required [16]. In addition, damage and contamination of a large-area MLG may occur in the two transfer processes, and economic loss follows. Therefore, to directly apply the rear MLG to the top electrode of the transparent OLED through a one-time transfer process, it is essential to develop a process that removes impurities from the rear MLG surface. In this study, we used pristine rear MLG (MLG) and rear MLG (O_2_-MLG) treated with oxygen plasma as top anodes in transparent organic light-emitting diodes (TOLEDs), and we report the effect of the plasma treatment on the electrical and optical properties of the devices. In addition, we introduce an analysis technology with limited probing depth to perform the surface-sensitive characterization.

## 2. Materials and Methods

Figure 1 shows the fabrication process of the inverted orange TOLED, composed of glass (0.7 mm), indium tin oxide (ITO, 70 nm), organic stack, and elastic graphene bonding structure (EGBS). The organic stack is composed of lithium (Li)-doped TRE (5%, 30 nm), TRE (25 nm), bis(2-phenylpyridinato)[2-(biphenyl-3-yl)pyridinato]iridium (III)(Ir(ppy)_2_(m-bppy))-doped PGH02 (8%, 20 nm), 4,4′,4″-tris(N-carbazolyl)-triphenylamine (TCTA, 10 nm), 1,1-bis(4-bis(4-methylphenyl)-amino-phenyl)cyclohexane (TAPC, 70 nm), and 1,4,5,8,9,11-hexaazatriphenylenehexacarbonitrile (HATCN, 30 nm). The EGBS consists of MLG [16] or O_2_-MLG, methacryloxypropyl terminated polydimethylsiloxane (DMS-R22), and poly-ethylene terephthalate (PET, 25 μm). Glass was used as a substrate. Then, ITO was used as a transparent bottom cathode, Li-doped TRE as an electron-injecting layer, TRE as an electron-transporting layer), (Ir(ppy)_2_(m-bppy))-doped PGH02 as an orange emissive layer, TCTA as an electron-blocking layer, TAPC as a hole-transporting layer, HATCN as a HIL, MLG or O_2_-MLG as transparent top anode, DMS-R22 as a bonding layer, and PET as a supporting layer were integrated in sequence to form functional layers.

The treatment of O_2_ plasma on MLG was performed for 30 s at a power of approximately 30 W, while the oxygen gas was flowed at approximately 100 sccm. The EGBS was transferred to the organic stack viaC gapless vacuum lamination in a glovebox [16]; then, TOLEDs were encapsulated utilizing an ultraviolet-curable epoxy and a glass cap containing a moisture absorbent material. In Figure 1, the laminating process consists of the following three steps [16]: the removal of nitrogen gas for 5 min at a vacuum pressure of 10^−2^ torr, maintaining a distance of about 2 cm between the EGBS sample and the organic stack; alignment of the two samples; and bonding to form a new HIL–MLG interface with a pressure of 110 kPa.

The current density (J)–voltage (V) characteristics of the TOLEDs with MLG and O_2_-MLG were measured using a Keithley 238 source measurement unit. The luminance (L) and electroluminescent (EL) spectra were measured using a spectroradiometer (CS-2000, Konica Minolta, Tokyo, Japan). The luminous current efficiency (η_LCE_) was calculated from the J-V-L characteristics and the EL spectra. The characteristics of top emission and bottom emission were respectively measured in the same pixel. The transmittance and reflectance of the TOLEDs were measured using an ultraviolet-visible infrared spectrophotometer (LAMBDA 750, PerkinElmer, Waltham, MA, USA) and their pixel size was 2 mm × 2 mm. The sheet resistance of the MLG was measured by a multimeter (Keithley 2000, Keithley Instruments, Inc., Cleveland, OH, USA) using a four-terminal measurement.

The electronic structures of the graphene surfaces were investigated using near-edge X-ray absorption fine structure (NEXAFS) spectroscopy at the 4B1 beam line in the Pohang Accelerator Laboratory (Pohang, Korea). The characterizations were carried out in a high vacuum system. For the analyses, specimens were attached to a molybdenum heatable holder and loaded into a vacuum chamber. In the NEXAFS analyses, carbon K-edge curves at photon energies of 274–324 eV were obtained from the partial-electron-yield (PEY) mode setup at photon incident angles of 30°, 45°, 55°, and 70°.

## 3. Results and Discussion

Figure 2a shows the J–V and L–V curves of the BE and TE OLEDs; the results are summarized in Table 1. The voltages (V_1000_) at approximately 1000 cd/m^2^ (L_1000_) of BE and TE O_2_-MLG devices are 12.0 and 12.6 V, respectively, which are much lower than those of the BE (13.8 V) and TE (15.3 V) MLG devices, respectively. Figure 2b shows the η_LCE_–J curves of the BE and TE OLEDs. η_LCE_ at L_1000_ of BE and TE O_2_-MLG devices are higher than those of the BE and TE MLG devices (MLG: 13.9 in BE, 13.0 in TE. O_2_-MLG: 15.7 in BE, 15.3 in TE). The results of V_1000_ and η_LCE_ indicate that the light-emitting characteristics of the MLG devices are significantly improved by the MLG surface treatment using oxygen plasma; these results are in good agreement with the previous report [18]. The maximum peak (λ_max_) of the EL spectra are similar for the four devices (553–554 nm, Figure 2c and Table 1) and sheet resistance of MLG does not change significantly before and after oxygen plasma treatment (MLG: 198 Ω/☐. O_2_-MLG: 212 Ω/☐). This indicates that the graphene anodes are not degraded by the surface treatment.

By applying front and rear light to the MLG-integrated devices (Figure 3a), optical property curves are obtained as a function of wavelength (λ) (Figure 3b,c). Interestingly, the MLG electrode itself showed a low transmittance of 77.2% [16], but the transmittance of the TOLED was quite high, over 70% at the wavelength of orange light (for example, the rear: 74.2% at λ = 554 nm; the front: 75.1% at λ = 553 nm). This means that TOLEDs can show the highest transparency when they have a single graphene electrode that exhibits near-perfect transmittance (99.7%) [22]. In contrast, OLEDs with metal anodes show semi-transparent or opaque characteristics. For example, the transmittance of the metal coated glass is reduced to less than half that of the blank glass substrate [23], and the value becomes lower when the organic stack and top electrode are formed on metal coated glass in fabricating OLEDs. Furthermore, in the wavelengths of 400–700 nm, the difference between the maximum and minimum transmittances of the graphene integrated devices is approximately 10%, which is much smaller than that difference for thin metal films (silver 35%, gold 35%, copper 15%) [24]. The difference between the maximum and minimum reflectance of graphene TOLEDs (approximately 17%) is thought to be smaller than those of the thin metals, because the sum of the transmittance and reflectance is nearly constant, indicating that uniform optical characteristics are obtained in the graphene devices: these characteristics are superior compared to those of thin metal integrated devices.

Angle-reserved carbon K-edge NEXAFS spectra of MLG samples are obtained from PEY mode setup at photon incident angles of 30°, 45°, 55°, and 70° (Figure 4a). The PEY spectra for measuring X-ray absorption are characterized by probing depths of 0.5–1.0 nm [25]. The resonance peaks at photon energies of 285.6 and 292.1 eV are formed by the electron transitions from carbon 1s to sp^2^ π* orbital or σ* of the C–C bond, respectively [26]. The peak at a photon energy of 287.8 eV originates from σ* bonds of hydrocarbon functional groups such as –CH, –OH, and –CO_x_, which favor the formation of defective graphene sites [27,28]. The broad curves above a photon energy of 300 eV are formed by the σ* resonances of the C=C bond [24]. As a non-dichroic response, in which the intensity of the π* peak does not depend on the incident angle, is observed at a photon energy of 285.6 eV, it is clear that MLG does not exhibit crystallinity within a probing depth of 0.5–1.0 nm. The molecular average tilt angle (α) between the molecular π-conjugated axis of the MLG and the surface plane is calculated using the Fermi golden rule equation and found to be 41° (Figure 4b) [29], also indicating that MLG is composed of only an amorphous phase. In particular, NEXAFS experiments using O_2_-MLGs (Figure 4c,d) show that α is 32°, indicating that the decrease of the angle from 41° (Figure 4b) to 32° (Figure 4d) by oxygen plasma treatment is due to the increased crystallization ratio. Furthermore, the oxygen treatment removes impurities in the graphene and thereby enhances the electrical properties of the anodes [30], indicating that the increased crystallization ratio and removal of impurities by oxygen treatment results in higher performance in O_2_-MLG devices compared with MLG devices. These findings indicate that the graphene surface plays an important role in enhancing device performance and that oxygen plasma treatment is a method that improves the electrical characteristic of the rear MLG, which is essential for developing next generation displays.

## 4. Conclusions

To explore the potential of using the rear face of graphene, TOLEDs using the rear faces of MLG and O_2_-MLG were fabricated, and the electrical and optical properties were characterized. O_2_-MLG devices had lower V_1000_ and η_LCE_ than MLG devices. In addition, the high transmittance and small difference between the maximum and minimum transmittance and reflectance provided the graphene-integrated TOLEDs with high transparency and uniform optical characteristics, which are essential to developing the transparent devices. NEXAFS spectra experiments indicated that the oxygen plasma treatment improved the light-outcoupling efficiency of the TOLEDs by increasing the crystallization ratio and removing impurities. We believe that this study provides insight into identifying specific strategies to regulate and tune the properties of surfaces/interfaces, which lead to highly efficient devices.

## Figures and Tables

**Figure 1 materials-14-06652-f001:**
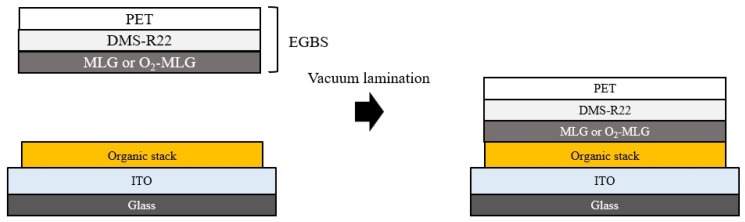
The fabrication of TOLEDs using vacuum lamination.

**Figure 2 materials-14-06652-f002:**
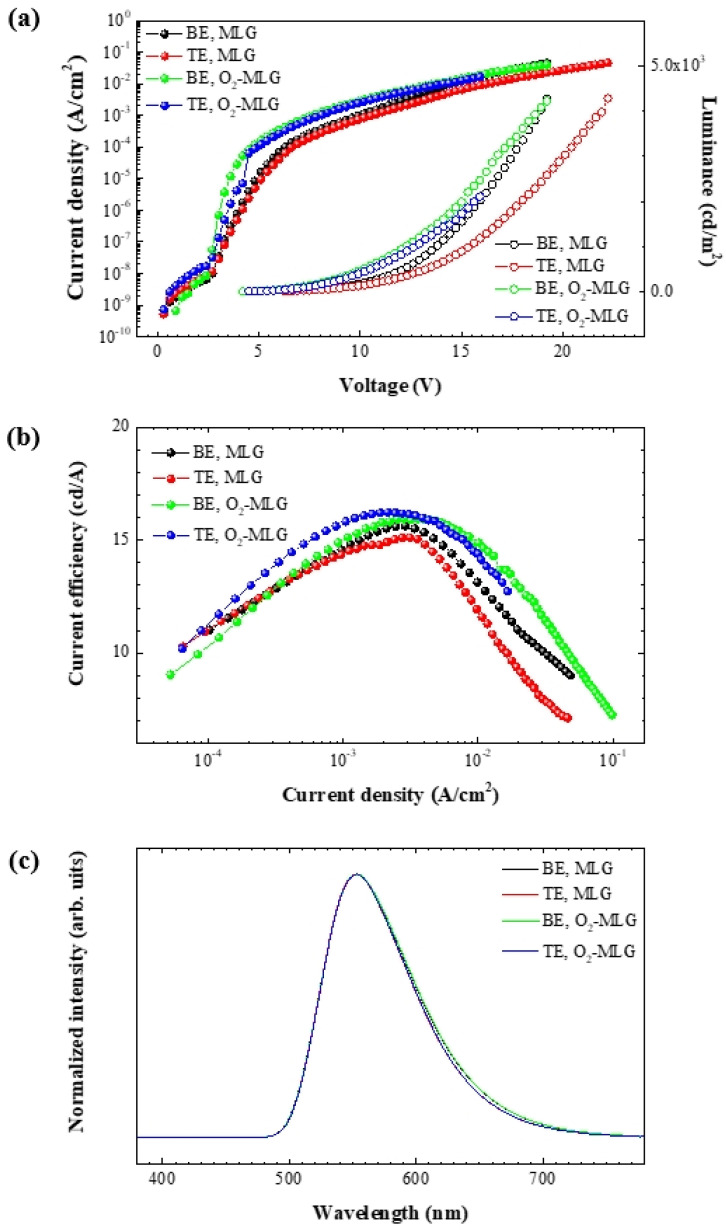
(**a**) J–V–L, (**b**) EL and (**c**) η_LCE_ characteristics of MLG and O_2_-MLG TOLEDs.

**Figure 3 materials-14-06652-f003:**
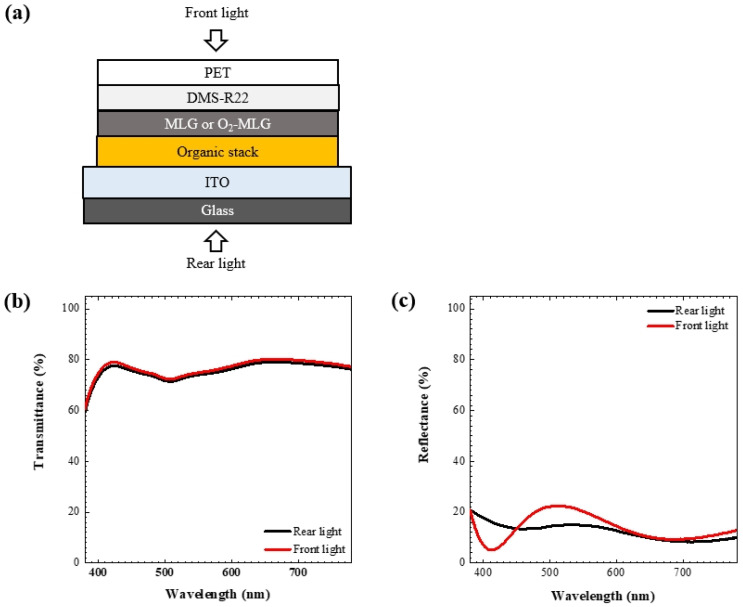
(**a**) Applying front and rear light to the graphene integrated devices. (**b**) Transmittance and (**c**) reflectance data, as a function of wavelength.

**Figure 4 materials-14-06652-f004:**
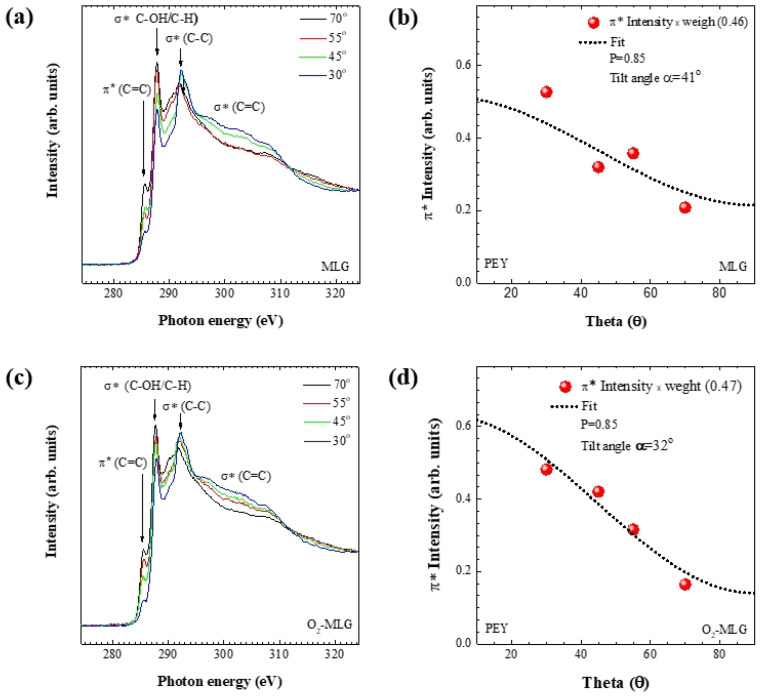
Angle-reserved carbon K-edge NEXAFS spectra of (**a**) MLG and (**c**) O_2_-MLG samples. The calculated molecular average tilt angles of (**b**) MLG and (**d**) O_2_-MLG samples, using the Fermi golden rule equation.

**Table 1 materials-14-06652-t001:** J–V–L characteristics of MLG and O_2_-MLG TOLEDs.

Anode	Emission Direction	V_1000_ at L_1000_ (V)	η_LCE_ at L_1000_ (cd/A)	λ_max_ at EL Spectra (nm)
MLG	BE	13.8	13.9	554
	TE	15.3	13.0	553
O_2_-MLG	BE	12.0	15.7	554
	TE	12.6	15.3	553

## Data Availability

The data presented in the article will be shared on request by the corresponding author.

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
