# Peer review of "Oxygen Plasma Treatment of Rear Multilayered Graphene: A Potential Top Electrode for Transparent Organic Light-Emitting Diodes"

_materials, 2021, doi:10.3390/ma14216652_

Round 1

Reviewer 1 Report

In this study, the rear faces of multilayered graphenes (MLG) without and with oxygen plasma treatment are used as top anodes to fabricate transparent organic light-emitting diodes. Results indicate that oxygen treatment significantly improves the light-emitting characteristics without degradation of the graphene materials.

General Comments

This work is clearly presented and interesting. However, the introduction should be improved and more references added. In particular recent reviews of this research field (e.g. Adetayo et al. Adv. Optical Mater. 2021, 2002102. DOI: 10.1002/adom.202002102). The beneficial effect of oxygen treatment in OLEDs is known (see e.g. Hwang et al. Appl. Phys. Lett. 2012   http://dx.doi.org/10.1063/1.3697639). Other studies concerning oxygen treatments should me mentioned to make clearer the contribution of this work to the field.The results are limited as the figures of merit of this type of devices are not discussed. The transmittances of the graphene devices are indicated as ≥ 74.2%. However, for industrial applications transmittances higher than 90% are necessary. It would be interesting to discuss means to improve reported results.

Specific comments

In Materials and Methods Section the source of the several materials used is not indicated. Several abbreviations in this section such as TRE and LGD225–doped PGH02 are not introduced in the text.

Reviewer 2 Report

The paper by Lim is a concise report on oxygen-plasma treatment of multilayered graphene for the use in TOLEDs. Electrical as well optical properties have been investigated.

Ref. 16 is a very detailed study by a group of expert scientist (and the author as first author) on face-dependent effects of multilayered graphene
in TOLEDs. The author may highlight this particular reference more prominently since there are many details on experiment and data analysis (such as how to apply Fermi's golden rule) as well as a thorough discussion given. The current submission that is rather brief will benefit from a more detailed description. 

Ref. 14: The name of the journal is "Society for Information Display, Digest of Technical Papers". 

Ref. 16: "Ihm, K." instead of "Lhm, K." (The same spelling error can be found in Ref. 8 but it has been published in that way.)

Ref. 23: Klyushin, A. Yu. (Russian Letter "Yu")

Ref. 27: Stöhr, J., NEXAFS Spectroscopy, Springer-Verlag Berlin, Heidelberg 1992. This book is written by a single author and part of the series (Vol. 25) "Springer Series in Surface Science".  

Reviewer 3 Report

The author reports oxygen plasma treatment of rear multilayered graphene for transparent organic light-emitting diodes. While the topic is interesting, the manuscript is short of convincing data to support itself. The following comments are provided.

  • For developing transparent electrodes, the measurements of conductivity and work function are required.
  • The manuscript shows high driving voltage for OLEDs based on rear multilayered graphene. The author is suggested reduce the driving voltage, for example, by inserting a thin layer of MoO3.
  • The English level of the entire manuscript needs to be improved.

Therefore, I cannot recommend it for the publication at the current form.

Reviewer 4 Report

The article reports about the use of rear multilayered graphene as transparent electrode in top emitting OLEDs. The author studied the effect of O2 plasma treatment on the characteristics of the transparent electrode in terms of OLEDs performances, optical and crystallinity properties and he observed an improvement in the performance of the device as a consequence of the plasma treatment. The development and the study of new materials as transparent electrodes is a topic of great interest in the field of flexible and performing optoelectronic devices and I think that this article can be accepted for publication after minor revisions:

  • the author should better explain the difference between front face (first born) and rear face (last born) MLG. I guess that the direct transfer of the rear face of the MLG on top of the organic stack is easier with respect the transfer of the front face, which includes two transfer processes of the multilayer structure so as to be able to expose the front face. I think that this aspect could be emphasized in the manuscript so as to demonstrate that the fabrication process is simpler than the one using the front face;
  • some material acronyms are not specified in the materials and methods section;
  • in the materials and methods section it is not specified the preparation method of the MLG;
  • about the O2 plasma treatment, in the materials and methods section it is not specified the employed O2 flow rate.
  • In figure 2, the efficiency curves (current efficiency or EQE) of the devices should be inserted;
  • The references list could be uploaded with more recent papers.

Reviewer 5 Report

The author of this work reported the rear face of MLGs without and with oxygen plasma treatment as top anodes of orange transparent organic light-emitting diodes (OLEDs). The optical characters were experimentally carried out. The structure and the writing of this manuscript are clear. However, the impact of this work is weak. Here are some of my comments.

  1. The author mentioned in the abstract " Very interestingly, the transmittances of the graphene devices are very high (≥ 74.2%).  " is not rigorous. Actually,  the transmittance is <74.2%. The relevant wavelength needs to point out when you want to discuss the transmittance.
  2. The comparison of the treated and untreated MLG transmittances need to be suplimented. 

Round 2

Reviewer 3 Report

The manuscript can be accepted for the publicaion.